# Update on the Pharmacological Treatment of Primary Biliary Cholangitis

**DOI:** 10.3390/biomedicines10082033

**Published:** 2022-08-20

**Authors:** Annarosa Floreani, Daniela Gabbia, Sara De Martin

**Affiliations:** 1Department of Surgery, Oncology and Gastroenterology, University of Padova, 35131 Padova, Italy; 2IRCCS Negrar, 37024 Verona, Italy; 3Department of Pharmaceutical and Pharmacological Sciences, University of Padova, 35131 Padova, Italy

**Keywords:** PBC, ursodeoxycholic acid (UDCA), obeticholic acid (OCA), fibrates, FXR agonists, PPAR agonists, budesonide

## Abstract

Ursodeoxycholic acid (UDCA) is the first-line therapy used for the treatment of PBC. In recent years, new pharmacological agents have been proposed for PBC therapy to cure UDCA-non-responders. Obeticholic acid (OCA) is registered in many countries for PBC, and fibrates also seem to be effective in ameliorating biochemistry alteration and symptoms typical of PBC. Moreover, a variety of new agents, acting with different mechanisms of action, are under clinical evaluation for PBC treatment, including PPAR agonists, anti-NOX agents, immunomodulators, and mesenchymal stem cell transplantation. Since an insufficient amount of data is currently available about the effect of these novel approaches on robust clinical endpoints, such as transplant-free survival, their clinical approval needs to be supported by the consistent improvement of these parameters. The intensive research in this field will hopefully lead to a novel treatment landscape for PBC in the near future, with innovative therapies based on the combination of multiple agents acting on different pathogenetic mechanisms.

## 1. Introduction

Primary biliary cholangitis (PBC) is a chronic liver disease characterized by autoimmune responses, in which the small interlobular bile ducts are progressively disrupted, causing cholestasis. As in other chronic liver diseases, PBC can evolve into hepatic fibrosis and cirrhosis, causing the need for liver transplantation to prevent liver failure and death [1]. In regard to its geographical distribution, the highest number of patients is diagnosed in Northern Europe and North America, even though this disease is also quite common in Europe (mainly in the Southern countries), Asia, and Australia. Its global prevalence is 14.6 per 100.000, and the global incidence is 1.76 per 100,000 per year [2,3]. A gender difference can be observed in PBC patients, with a female predominancy. A F/M ratio of 9:1 was reported in a cohort series analyzing epidemiology, natural history, and clinical characteristics of PBC patients [1].

The clinical features and natural evolution of PBC may vary greatly between patients, who can experience either asymptomatic, slowly progressive, symptomatic, or rapidly evolving disease. Over the last 30 years, a modification of PBC symptoms was observed, which has changed from a disease with evident clinical manifestations, such as portal hypertension, to a milder condition, characterized by a long outcome [4]. The etiology of PBC is complex, and some mechanistic issues remain to be solved. Nevertheless, there is a general consensus indicating a predisposing genetic background that could lead to the onset of the disease in combination with infective, immunological, and/or environmental triggers [5,6,7]. The therapeutic management of PBC is a fascinating challenge, and several drugs with different mechanisms of action are either approved or under development (Figure 1).

## 2. Ursodeoxycholic Acid (UDCA) as First-Line PBC Therapy

The standard therapy for PBC is currently ursodeoxycholic acid (UDCA), a natural hydrophilic tertiary bile acid with choleretic properties, used in clinical practice at the dose of 13–15 mg/Kg per day, according to the European guidelines. The mechanism of action of UDCA is complex and involves several molecular pathways, which have been extensively studied in preclinical settings. There is a general consensus that its therapeutic effect on PBC is mainly due to: (1) the stimulation of hepatocellular secretion and (2) the stimulation of cholangiocellular secretion, both resulting in a choleretic effect; (3) anti-apoptotic effects on hepatocytes; (4) the reduction of bile acid toxicity. UDCA exerts its mechanisms of action by interacting with a panel nuclear receptors, i.e., retinoid X receptor (RXR), peroxisome proliferator-activated receptor α (PPARα), and pregnane X receptor (PXR), all of which transcriptionally modulate the synthesis and homeostasis of bile components [8]. The drug is given as a single dose, or divided into multiple doses, due to tolerability issues [9]. Observational studies evaluating PBC patients treated with UDCA helped to define an achievement of biochemical response in therapy-responding patients, evaluating also the prolongation of liver transplant (LT)-free survival, with respect to non-responders. Altogether, the data collected with these clinical studies helped to define a population of PBC patients in which UCDA therapy is beneficial [10,11,12,13,14,15]. A multicentric study evaluating PBC patients treated with UDCA or placebo demonstrated that plasma bilirubin values below the current upper limits of normal (ULN) are predictive of survival, and a threshold of 0.6 x ULN was selected for assessing the increased risk of LT or death [16]. Furthermore, a study observed that a relevant proportion of PBC patients has an incomplete biochemical response to UDCA therapy, according to the Paris II criteria, and the presence of cirrhosis, elevated GGT, and alkaline phosphatase (ALP) at diagnosis could represent predictive factors for an incomplete UDCA response [17]. However, UDCA therapy has been demonstrated to improve LT-free survival in all PBC patients, irrespective of diseases severity and whether or not they meet the accepted criteria for the definition of a UDCA responder [18]. Thus, these observations are likely to prove that the improvement of cholestatic biochemical parameters in PBC patients, even of modest entity, can generate long-term benefits. However, despite all these efforts, the correlation between the lack of UDCA efficacy and survival in PBC patients still needs to be defined in detail. Two groups, i.e., the Global PBC Study Group and the United Kingdom (UK)-PBC Consortium, have been created with the aim of setting up a prognostic model for disease progression in UDCA-treated patients. These two groups independently developed and evaluated the risk of PBC progression. In 2015, the first score, called the GLOBE score, was introduced to assess PBC risk progression. The setting up of this score accounted for a wide derivation cohort (accounting for 2488 patients) and a validation cohort accounting for 1634 UDCA-treated patients. In the same years, another score was proposed by the United Kingdom (UK)-PBC Consortium, called the UK-PBC risk score (www.uk.pbc.com (accessed on 15 July 2022)), based on a nation-wide cohort of 1916 patients and validated in a cohort of 1249 UDCA-treated patients. These two predictive models have also been validated in PBC subjects not treated with UDCA, providing indications of disease activity and stage, based on biochemical liver function markers. The two main differences between the two models rely on the different endpoints used for calculating the scores. First, the GLOBE-PBC score takes into account all-cause mortality, in addition to LT-related mortality, whereas the UK-PBC score considers only liver-related death. Interestingly, in the study population of the Obeticholic Acid International Study of Efficacy (PBC POISE), both models demonstrated a potential usefulness in individualizing risk prediction, both in clinical practice and therapeutic trials for PBC [19]. It should be noted that the UDCA non-responder patients are around 30–40% of all UDCA-treated patients. Since they have a poorer prognosis due to a higher risk of disease progression, and the plausibility to require liver transplantation, as well as a greater mortality risk [20], the identification of novel effective treatments still represents an urgent medical need.

## 3. Other Therapeutic Agents for PBC

To overcome the problem of the incomplete response to UDCA and/or toxicity issues, several alternative therapeutic approaches have been proposed, and many clinical trials are currently ongoing to assess the possibility of repositioning approved drugs after the demonstration of their efficacy in PBC patients. Furthermore, a variety of candidate drugs are under evaluation in clinical trials for PBC patients because of their promising mechanisms of action, i.e., bile acid modulation, immunomodulation, and antifibrotic and anti-inflammatory effects. Table 1 summarizes the ongoing clinical trials.

### 3.1. Obeticholic Acid

The only second-line drug approved for the treatment of PBC is obeticholic acid (OCA), which is indicated for patients who are non-tolerant or non-responding to UDCA after 12 months of treatment. OCA is a chemically modified derivative of BA chenodeoxycholic acid. Its mechanism relies on an agonistic activity on the farnesoid X receptor (FXR). Thanks to its affinity to FXR, OCA regulates the synthesis and export of bile acids (BAs), thereby preventing hepatic toxicity due to their toxic accumulation [21]. Beside the regulation of BA homeostasis, its complex and multifaced mechanism of action comprises anti-inflammatory and antifibrotic effects, as demonstrated by preclinical and clinical studies [21,22], thereby targeting a panel of pathological processes involved in PBC development.

The first clinical indication for the use of OCA in monotherapy came from an international randomized, double blind, placebo-controlled phase 2 study investigating the benefit of treating PBC patients with OCA in monotherapy [23]. In this study, patients were randomized into three groups, i.e., 23 PBC patients treated with placebo, 20 with OCA (10 mg dose), and 16 patients with OCA (50 mg dose) for 3 months. As a primary endpoint, the ALP percentage change from baseline was considered. OCA significantly reduced ALP levels in patients treated at both dosage with respect to placebo. This study also reported an improvement in many biochemical parameters, among which were GGT, alanine aminotransferase, conjugated bilirubin, and immunoglobulins. The most common adverse drug reactions (ADRs) observed after OCA therapy was pruritus, which was reported in this study in patients treated with both OCA 10 mg (15%) and 50 mg (38%).

The FDA approved OCA in 2016 after the results of the phase 3 international trial POISE, with a multicentric randomized controlled design, enrolling more than 200 PBC patients [24]. Interestingly, it should be emphasized that more than 50% of UDCA-non-responders had a beneficial effect by receiving the combination therapy of OCA plus UDCA for 12 months, as indicated by the achievement of the clinical endpoint, which was an ALP level lower than 1.67 times ULN, reduced by at least 15% from baseline. After 12 months, all patients received OCA therapy in the extension phase [25]. In the following 3-year interim analysis, OCA obtained good results on both efficacy and safety, demonstrating a stable therapeutic performance, even associated with a significant reduction in total and direct bilirubin, more evident in patients with high baseline direct bilirubin [26], and good tolerability. The most common adverse drug reactions (ADRs) reported in the POISE trial were pruritus and fatigue, which were experienced by 77% and 33% of OCA-treated patients, respectively [26]. Pruritus received the score “mild-to-moderate” by the visual analogue scale (VAS), and 8% of patients (n = 16) had to withdraw due to this ADR. However, most patients experiencing severe pruritus have been treated with specific drugs. The histological analysis of a subgroup of 17 patients recruited in the POISE trial who underwent liver biopsy at the time of enrollment and after 3 years of treatment, showed that the chronic therapy with OCA led to an improvement or at least a stabilization of the histology of PBC patients, assessed by evaluating ductular injury, fibrosis, and collagen morphometry [27]. Another sub-analysis of patients enrolled in the POISE trial investigated the beneficial effects of OCA on AST to platelet ratio (APRI) and transient elastography (TE), which are both non-invasive markers of liver fibrosis [28]. A significant APRI reduction from the baseline could be observed in OCA-treated patients and during the open-label extension phase with respect to placebo-treated group. Furthermore, the treatment with OCA 10 mg caused a decreasing tendency toward liver stiffness, while both patients treated with lower dosages of OCA or placebo showed a mean increase in liver stiffness [28]. Despite the small sample size, this study can be considered as a milestone in PBC therapy, since it demonstrated that most patients who respond inadequately to UDCA ameliorated or stabilized multiple histological PBC features when treated with OCA.

The decision to implement PBC pharmacological therapy with OCA deserves consideration if at least one of the following conditions is met: (i) ALP ≥ 1.67 x ULN (in Italy, the ALP threshold is 1.5 ULN); (ii) total bilirubin > ULN, but < 2 x ULN.

Three clinical studies analyzing real-world cohorts of OCA patients have been published so far, all reporting results for 12 months of OCA treatment [29,30,31]. The first real-world analysis on the effectiveness of OCA treatment was conducted on 64 Canadian PBC patients experiencing incomplete UDCA response, or who were intolerant to UDCA [29]. Among the 44 patients meeting the inclusion criteria of POISE, 39% (n = 17) underwent a 1-year biochemical evaluation. While only 18% of these patients (3 out of 17) reached the POISE primary endpoint after 12 months of treatment, 43% of patients (9 out of 21) achieved this target after a 19-month observation period. Overall, a significant ALP, GGT, transaminases, and IgM reduction was reported in the whole cohort. As regarding pruritus, either new onset or exacerbation was reported in 26 patients (41%), and 5 of them had to discontinue the drug for this reason. Other reasons for therapy discontinuation reported in this cohort were skin rash (n = 2), liver toxicity (n = 2), and incomplete response after 12 months of treatment (n = 2). In the Iberian cohort [30], 120 patients were enrolled (21.7% of them had cirrhosis and 26.7% received or were taking concomitant treatment with fibrates). A total of 78 patients completed at least 1 year of treatment. The GLOBE-PBC score significantly decreased to 0.17 (*p* = 0.005), whereas the UK-PBC score decreased to 0.17, without reaching any significant difference (*p* = 0.11). According to the POISE criteria, 29.5% of patients achieved a response. In the Italian cohort recruited into the Italian PBC registry, 191 patients were analyzed [31], and 43% of them responded to OCA, according to the POISE criteria. Patients with cirrhosis showed lower efficacy (29.5%). Patients with AIH/PBC overlap syndrome showed a comparable efficacy to classical PBC, with a higher ALT reduction at 6 months. A further analysis was conducted in 100 cirrhotic patients from the Italian cohort (De Vincentis A, unpublished). The response to treatment, according to the POISE criteria, was obtained in 41% of cases, also confirming the efficacy of the drug in cirrhotic stage. Of note, by applying the normal range criteria, 11.5% of the cirrhotic patients reached the endpoint. A total of 22 patients (22%) discontinued the treatment due to severe side effects (5 patients with jaundice and/or ascitic decompensation and 4 with upper digestive bleeding. One patient died after TIPS placement).

### 3.2. Non-Bile Acid FXR Agonists

Three compounds without the classical bile acid structure, but able to bind and activate FXR, have been proposed to treat PBC patients, i.e., tropifexor, cilofexor, and EDP-305.

Tropifexor is a highly potent FXR agonist with a positive effect in treating both cholestasis and steatosis in animal models, mainly by reducing fibrosis [32]. A phase 2 study investigated tropifexor efficacy in PBC patients characterized by an inadequate UDCA response. Patients were randomized in arms, receiving once daily doses of 30 μg, 60 μg, or 90 μg of tropifexor or placebo for 4 weeks [33]. Moreover, an interim analysis was conducted in the cohort of patients treated with 90 μg. In this group of patients, a rapid decrease in the levels of GGT (72% reduction), ALP, ALT, and AST could be observed at day 28, as well as a good tolerability of tropifexor. HDL was reduced by 33% and 26% at the doses of 60 and 90 μg, respectively, and restored to physiological levels by the end of the study. No increase was observed in total or LDL cholesterol. The results of this trial suggested that this FXR agonist is a candidate drug for PBC therapy [33].

Another non-steroidal FXR agonist, called cilofexor, was tested in a trial (NCT02943447) enrolling 71 UDCA non-responders with PBC. They were randomized into three groups treated with 30 or 100 mg cilofexor or placebo once a day for 12 weeks. Patients treated with 100 mg cilofexor achieved a significant median reduction in GGT (8–47.7%, *p* < 0.001), ALT (8–13.8%, *p* = 0.05), C-reactive protein (8–33.6%, *p* = 0.03), and primary bile acids (−30.5%, *p* = 0.008). The reduction in ALP was greater than 25% in 17% of the patients treated with the dose of 100 mg and in 18% of those treated with 30 mg cilofexor, in comparison with 0% obtained in the placebo group. The major ADR observed after cilofexor treatment was pruritus, particularly common in patients treated with the higher dose. Moreover, promising results were obtained from a phase 3 trial in patients with PSC, thus suggesting the potential benefit of using this new non-bile acid FXR agonist [34].

EDP-305 is a potent FXR agonist with a “mixed” structure, containing steroid and non-steroid moieties, without the classical carboxylic acid group of the other FXR agonists or natural bile acids. The INTREPID study (NCT03394924) evaluated its safety, tolerability, and efficacy in PBC patients with inadequate response to UDCA. A total of 68 subjects were randomized to receive either EDP-305 2.5 mg, 1 mg, or placebo for 42 weeks [35]. The primary endpoint was the proportion of patients with at least 20% ALP reduction from the pre-treatment value, or normalization of ALP at week 12. The intention-to-treat analysis showed that EDP-305 resulted in ALP reduction of 45% and 46% in the 1 mg and 2.5 mg treatment groups, respectively, whereas this reduction was only 11% in the placebo group. Five patients treated with 2.5 mg EDP-305 had severe pruritus. Pruritus was present in 51% of the 2.5 mg-treated patients, whereas less than 10% of patients treated with 1 mg experienced it. In general, the other most common ADRs were gastrointestinal-related symptoms, e.g., nausea, vomiting, diarrhea, or headache, and dizziness.

### 3.3. PPAR Agonists: Fibrates

Fibric acid derivatives, also called fibrates, are an old class of lipid-lowering agents proposed as a second-line PBC therapy in the late 1990s. The first drug belonging to this class was clofibrate, discovered in 1962 [36]. These drugs attracted great attention for treating PBC patients because they showed efficacy against cholestasis, inflammation, and fibrosis. Their mechanism of action relies on their agonist effect on peroxisome proliferator-activated receptors (PPARs), a family of nuclear receptors (NRs). Three main isoforms of PPARs have been described, i.e., α, β/δ, and γ, each encoded by distinct genes and characterized by a peculiar tissue distribution. Each fibric acid derivate displays a peculiar pattern of affinity towards these three PPAR isoforms, thus differently modulating PPAR-related pathways. Fenofibrate, by binding to PPARα, stimulates the transcription of the multidrug resistance protein 3 (MDR3) transporter, increasing the biliary secretion of phosphatidylcholine and improving cholestasis biomarkers [37]. At variance, bezafibrate, beside binding to PPARα and γ, is also an agonist of pregnane X receptor (PXR) [38], another transcription factor implicated in cholestatic liver disease [39]. The first placebo-controlled trial investigating the efficacy of fibrates in PBC treatment was the BEZURSO trial, a phase 3 study proposing a combination therapy with bezafibrate (BEZA) and UDCA. This study demonstrated that the addition of BEZA to the previous monotherapy of UDCA induced a significantly higher biochemical response with respect to patients of the placebo/UDCA arm [40]. This result was also associated with an improvement in PBC symptoms and surrogate markers of fibrosis. The main ADRs associated with the use of fibrates were linked to creatinine and transaminase increase and heartburn. In addition, gallstone formation, perhaps as consequence of the reduction in BA synthesis, and a paradoxical increase in cholesterol, have also been reported in PBC patients treated with clofibrate, even though the same ADRs have not been observed in patients treated with fenofibrate (FENO) or bezafibrate [41].

To compare the efficacy of OCA and fibrates as second-line therapies, a multicentric retrospective study including PBC patients from 30 centers has been undertaken in Spain [42]. A total of 86 patients receiving OCA (5 mg), 250 patients receiving fibrates (81% BEZA 400 mg, 19% FENO 200 mg), and 15 receiving OCA plus fibrates were enrolled. Both treatments decreased GGT and transaminases and improved the GLOBE score. ALP decrease was higher in patients treated with fibrates, whereas alanine aminotransferase was lower in OCA-treated patients. Discontinuation was more frequent in fenofibrate treatment due to low tolerability or the onset of ADRs. In summary, neither OCA nor fibrates emerged as a significantly better second-line treatment for PBC. Caution should be recommended, in any case.

At the AASLD meeting in Boston in 2019 [43], the results of another trial assessing the comparative efficacy of BEZA or OCA in 59 patients was presented. This study did not reveal significant differences in the incidence of severe hepatic impairment manifestations, such as varices, variceal bleeding, ascites, and LT list insertion between patients treated with OCA or bezafibrate. However, ALP reduction was more evident in bezafibrate-treated patients with respect to those treated with OCA (*p* < 0.001). A higher percentage of BEZA-treated patients experienced an elevation of bilirubin. These two studies offer great insight by presenting real-world data regarding the use of OCA and fibrates in PBC patients, paving the way for the design of future trials.

The additive effects of the combination of fibrates and OCA were investigated in a multicenter retrospective cohort of 58 patients with PBC [44]. A total of 50 of them were treated with a combination of OCA (5–10 mg/day), fibrates (BEZA 400 mg/day or FENO 200 mg/day), and UDCA (13–15 mg/day). Triple therapy was associated with a significant decrease in ALP levels with respect to dual therapy, and with an odds ratio for ALP normalization of 5.5 (95% CI: 1.8–17.1, *p* = 0.003).

Regarding the effect of fibrates on pruritus, this deserves a separate discussion. The benefit of fibrates in improving this symptom is well documented. The Fibrates for Cholestatic Itch (FITCH) trial was designed to investigate the effects of BEZA on pruritus in 70 patients with PBC, primary sclerosing cholangitis (PSC), or secondary sclerosing cholangitis who reported pruritus scored as “moderate to severe” [45]. The primary endpoint of this trial was the achievement of a reduction of more than 50% of VAS-assessed pruritus. BEZA (400 mg/day) led to this achievement in 45% of patients (41% PSC, 55% PBC), whereas only 11% reached the primary endpoint in the placebo group (*p* = 0.003). This effect in relieving cholestasis-associated pruritus occurs via an autotaxin-independent mechanism [46]. This improving effect on pruritus ensures that fibrates should be employed as a second-line option for PBC therapy in patients experiencing moderate to severe pruritus.

Since fibric acid derivates reduce cholesterol levels, they should be considered for the treatment of PBC patients with hypercholesterolemia associated with low levels of high-density lipoprotein [HDL], in whom these agents are protective against cardiovascular events.

### 3.4. Other PPAR Agonists

The efficacy of elafibranor (ELA), an agonist of PPAR α and δ, has been recently investigated in PBC patients enrolled in a phase 2, double-blind, placebo-controlled study [47]. A total of 45 PBC patients with inadequate UDCA response were randomized into three groups, receiving either ELA 80 mg or ELA 120 mg four times a day, or placebo four times a day for 12 weeks (NCT03124108). ELA significantly decreased mean ALP at week 12 in both groups (−48% in 80 mg-treated group and −40.6% in 120 mg-treated group, *p* < 0.001). The endpoint (ALP < 1.67 x ULN, ALP decrease >15%, and total bilirubin < ULN) was reached in most (67% and 79%) patients treated with the 80 or 120 mg doses, respectively. Moreover, in ELA caused an improvement in lipid and inflammatory markers (IgM, CRP, haptoglobin, fibrinogen) and a decrease in 7α-hydroxy-4-cholesten-3-one, or C4, an intermediate of bile acid synthesis. ELA at both dosages was well tolerated and did not cause induction or exacerbation of pruritus. In general, all these effects suggest that ELA is a promising drug candidate for PBC.

A 12-week double-blind, randomized, placebo-controlled phase 2 trial investigated the effect of seladelpar, a selective PPAR-δ agonist [48]. A total of 70 PBC patients with inadequate response or intolerance to UDCA were randomly divided into 3 experimental groups treated with 50 or 200 mg/day of seladelpar or placebo. Since 3 patients treated with seladelpar developed a grade 3 increase in ALT, even if fully reversible and asymptomatic, the study was terminated early. Despite these results, the safety and tolerability of seladelpar have been tested in a 52-week, phase 2, open-label uncontrolled dose-finding study [49,50]. This trial enrolled 120 patients who were treated for 12 weeks: 53 patients were treated with seladelpar 5 mg/day, 55 with seladelpar 10 mg/day, and 11 were assigned to the 2 mg/day group (United Kingdom sites after interim analysis), after which the dose could be increased to 10 mg/day. One year of observations indicated that seladelpar appeared to be safe and well-tolerated, while not inducing pruritus. A total of 4 patients discontinued seladelpar due to ADRs, 2 of which have been correlated to the drug treatment (grade 1 heartburn and grade 2 transaminase elevation). The composite endpoint (ALP < 1.67 x ULN, −15% reduction in ALP, total bilirubin < ULN) was met in 64% and 67% of seladelpar-treated patients. ALP normalization rates were 9%, 13%, and 33% in the 2 mg-, 5 mg-, and 10 mg-treated groups, respectively. After one year of treatment with seladelpar, 101 patients included in this trial self-reported using the pruritus VAS, the 5D-itch scale, and the PBC-40 questionnaire (evaluating itch and fatigue domains) [51]. Seladelpar led to consistent improvement in both pruritus and fatigue, along with a reduction in serum bile acid profiles. A phase 3, international, randomized, placebo-controlled study (ENHANCE) further assessed the safety and efficacy of seladelpar in PBC patients not responding to first-line treatment [52]. Enrolled participants were randomized into three groups of 80 patients: seladelpar 10 mg/day, seladelpar 5 mg/day, or placebo. After a first analysis after 26 weeks, patients were treated for an additional 26 weeks with either 5 mg or 10 mg of seladelpar. The primary endpoint was a composite response at month 3, which included an ALP of less than 1.67 times the ULN, a ≥15% ALP reduction, and total bilirubin at or below the ULN. After one year of treatment, this study demonstrated a mean ALP decrease of 40% in the 5/10-mg group and of 45% in the 10-mg group. In addition, in the 5-mg group uptitrated to 10 mg, 53% of the patients reached the composite endpoint, as well as 69% of patients in the 10 mg group. However, this trial was terminated early due to an unexpected histological finding of non-alcoholic steatohepatitis, even though the causality assessment with seladelpar treatment was not demonstrated. These results suggest that seladelpar is a drug candidate for the second-line therapy of PBC, although further evidence about its tolerability should be obtained.

### 3.5. Agents Targeting the FGF19 Pathway

Fibroblast growth factor 19 (FGF19) is a hormone encoded by the *FGF19* gene, directly reducing the gene expression of CYP7A1, a key enzyme catalyzing the first rate-limiting step of bile acid synthesis [53]. Since the suppression of hepatocyte bile acid synthesis is a rational mechanism for the improvement of bile acid homeostasis and the management of cholestasis, some attempts to find novel agents acting via the FGF19 axis have been made.

An engineered analogue of FGF19, NGM282 (aldafermin), was tested in a multicentric, randomized, double-blind phase 2 trial [54]. A total of 45 PBC patients with inadequate UDCA responses were randomly assigned to three groups: one received subcutaneous daily administration of NGM282 at a 0.3 mg dose (n = 14), another received 3 mg (n = 16), and the latter received the placebo (n = 15). NGM282 treatment significantly reduced ALP (primary endpoint) at both doses compared to placebo at the end of the treatment. Moreover, 50% of the patients receiving 0.3 mg and 46% of those receiving 3 mg were shown to have an ALP reduction higher than 15% from baseline compared to 7% in the placebo group. Most ADRs were gastrointestinal disorders of grade 1 and 2. Overall, the tolerability profile of NGM282 was acceptable. However, further studies are encouraged to ascertain whether the biochemical response is durable and related to a real improvement of robust clinical outcomes, rather than to a decrease in decompensation or death.

### 3.6. Agents Targeting the NADPH Oxidase (NOX) Enzymes

Besides their physiological functions, NADPH oxidases (NOXs), enzymes devoted to the production of reactive oxygen species [55], play a role in multiple pathological processes characterized by excessive oxidative stress. The NOX inhibitor GKT831 (setanaxib) was investigated in a phase 2 trial including 111 patients with PBC divided into 3 arms, one receiving 400 mg of GKT831 once daily (n = 38), another twice daily (n = 36), and the latter receiving placebo (n = 37) [56]. The primary endpoint was the change in GGT vs. baseline, and the secondary endpoints were the modification in ALP, liver stiffness evaluated by means of FibroScan, and overall quality of life after 24 weeks. GKT831 led to a reduction in cholestatic markers. Particularly, a greater GGT reduction was observed in patients with higher baseline values, thus suggesting that this NOX inhibitor may be useful in patients with more advanced disease. Moreover, GKT831 was shown to be well-tolerated, with no reported treatment discontinuation or interruption due to pruritus or fatigue. Due to the positive results obtained in this trial, a phase 3 trial in PBC patients is planned.

### 3.7. Agents with Immunomodulatory Properties

In recent years, many studies have pointed out that immunomodulators, such for example anti-IL antibodies, Janus kinase (JAK) 1/2 inhibitors and sphingosine-1-phosphate receptor agonists, may have a potential efficacy in the treatment of PBC, since the dysregulation of innate immune system plays a fundamental role in its pathogenesis. To date, some agents with immunomodulatory properties are in early-stage preclinical and/or clinical development for PBC treatment.

Budesonide, a synthetic corticosteroid displaying a high first-pass metabolism, has been evaluated in a placebo-controlled, double-blind trial in 62 non-responder patients to UDCA [57]. Participants were randomly assigned 2:10 to receive budesonide (9 mg/day) or placebo once daily for 36 months while maintaining UDCA treatment. The primary endpoint was the improvement of liver histology with respect to inflammation and no progression of fibrosis. Comparing patients with paired liver biopsies (n = 43) the histologic endpoint was not met; moreover, serious adverse events occurred in 10 patients receiving budesonide and 7 receiving placebo. Improvements in biochemical markers of disease activity were obtained with budesonide.

Recently budesonide has been recommended for patients diagnosed with AIH/PBC overlap syndrome [58]. This treatment can improve liver function tests and is relatively safe, although the risk of portal vein thrombosis remains a concern [59].

The efficacy of rituximab, an anti-CD20 chimeric monoclonal antibody, was evaluated in two open-label studies enrolling PBC patients with incomplete UDCA response. The results of both studies suggested a limited efficacy of rituximab in PBC patients, even though an impressive reduction in ALP levels was observed [60,61] in a limited number of patients. Moreover, the treatment with rituximab was demonstrated to be ineffective in reducing fatigue in a phase 2 trial in PBC patients [62].

Since PBC hepatic histology shows a lymphocytic infiltration in portal tracts and segmental inflammatory destruction of intrahepatic bile ducts, some studies have investigated the potential effects of antibodies directed against chemokine (C-X-C motif) ligand 10 (CXCL10) in PBC patients. CXCL10 is a chemokine secreted in response to interferon-γ-stimulation by several cell types, e.g., monocytes, endothelial cells, fibroblasts, cholangiocytes, and hepatocytes, and is implicated in the hepatic recruitment of inflammatory T cells. This effect is elicited through its binding to chemokine (C-X-C motif) receptor 3 (CXCR3), expressed on effector T cells [63]. Moreover, both CXCL10 and CXRC3 have been demonstrated to be upregulated in the serum and livers of PBC patients [64]. In particular, CXCR3+ cells have been found in the hepatic tissue of PBC patients [65]. Interestingly, in situ hybridization of PBC liver samples demonstrated the presence of the CXCL10 messenger in hepatocytes surrounded by infiltrating monocytes. The anti-CXCL10 monoclonal antibody NI-0801 was evaluated in a phase 2 study enrolling 29 UDCA-non-responder patients with PBC [66]. Each patient received an intravenous infusion of NI-0801 (10 mg/Kg, 6 doses) every 2 weeks. A 3-month follow-up was performed after the last infusion. No serious ADRs have been reported after treatment, and the most common ADRs were headaches (52%), pruritus (34%), fatigue (24%), and diarrhea (21%). However, the trial was terminated due to no significant therapeutic benefits obtained, despite the good pharmacological response observed in the blood, since the high rate of CXCL10 production makes it difficult to reach drug levels leading to an effective sustained neutralization of this chemokine [66].

Ustekimumab, a monoclonal antibody specifically binding the two interleukins IL-12 and IL-23, has been investigated in a multicentric, open-label study including PBC patients with an inadequate response to UDCA. Unfortunately, the results of this study failed to demonstrate the efficacy of this antibody in achieving a decrease, even moderate, in ALP levels [67].

Another open-label trial investigating abatacept, a fusion protein formed by the extracellular domain of the CTL4 and Fc region of the immunoglobulin IgG1, has demonstrated the inefficacy of this protein in achieving the required clinical outcomes [68].

Baricitinib is a JAK inhibitor, selective for the two subtypes JAK1 and JAK2, already approved in the US and Europe for the treatment of other autoimmune diseases, e.g., rheumatoid arthritis, and alopecia areata. JAK is a family of intracellular tyrosine kinases transducing cytokine-mediated signals. A randomized, double-blinded placebo-controlled trial in patients with PBC and inadequate response or intolerance to UDCA was performed [69]. Endpoints included change in ALP, itch numeric rating score, and fatigue scoring at 12 weeks post-baseline. Only two patients were enrolled and completed the trial (one received baricitinib and the other placebo). The patient treated with baricitinib demonstrated a 30% decrease in ALP and a 7-point improvement in itch scoring, but a 2-point increase in fatigue scoring.

FFP-104, an anti-CD40 monoclonal antibody, is a novel agent proposed for the treatment of PBC, since CD40 promotes the efficient T cell activation caused by the paracrine communications of antigen presenting cells, fibroblasts, and other non-lymphoid cells. As a consequence, its blockade could be exploited to counteract PBC autoimmune activation. A phase 2 trial including PBC patients is currently ongoing to determine the initial safety, tolerability, and pharmacodynamics of this antibody in PBC patients (NCT02193360). Interestingly, in a murine model of autoimmune cholangitis, administration of the anti-CD40 ligand reduced liver inflammation and lowered the levels of AMA, but these reductions were not sustained [70].

Mesenchymal stem cells (MSCs) transplantation has been studied as alternative to liver transplantation for patients with end-stage PBC [71]. MSCs are able to modulate and repair the injured tissue by affecting immune response by different mechanisms, such as cell-to-cell interactions and the production of useful paracrine factors [72]. The first clinical trial evaluating MSCs for PBC treatment was conducted in China (NCT01662973). This pilot study enrolled a small number of patients (n = 7) with an incomplete response to assess the safety and efficacy of umbilical cord-derived mesenchymal stem cells (UC-MSCs) [73]. All patients received 3 infusions of UC-MSCs every 4 weeks. After 48 weeks of follow-up, the treatment was well tolerated, and no relevant ADRs occurred. UC-MSCs significantly decreased serum levels of ALP and GGT. After these encouraging results, a second study was performed by the same research group testing MSCs derived from allogenic donors of the patients’ family members [74]. Their efficacy was evaluated using a 1-year of follow-up. Although transaminases, GGT, and IgM were significantly improved, the histological analyses evaluating the presence and severity of fibrosis were stabilized by the treatment. Overall, further studies seem to be necessary to discriminate the real efficacy of the use of MSC therapy in PBC. A new study is currently ongoing (NCT03668145).

### 3.8. Antiretroviral Therapy

After the proposal of a Canadian research group of a viral involvement in the pathogenesis of PBC, a multicentric trial was designed to investigate the efficacy of antiretroviral therapy in PBC patients (NCT01614405) in a limited number of patients (n = 13), since most enrolled patients were intolerant to the lopinavir-ritonavir (LPRr) combination. Patients were randomized and received a combination of tenofovir-emtricitabine (TDF/FTC 300/200 mg), LPRr (800/200 mg), or placebo for 6 months [75]. A significant 25% reduction in ALP was observed after antiretroviral therapy (*p* < 0.05). However, an important limitation to the use of the antiviral combination was represented by the frequency of ADRs, which were much higher than those reported in HIV patients receiving the same therapy. A new trial investigating better-tolerated combination regimens is ongoing (NCT03954327). Another antiretroviral therapy with tenofovir/emtricitabine-based regimens in combination with lopinavir or raltegravir in recurrent PBC following liver transplantation improved hepatic biochemistry, but the antiretroviral therapy was associated with side-effects [76].

## 4. Agents for the Treatment of Specific Symptoms of PBC

### 4.1. Agents Targeting Pruritus

Pruritus represents a frequent and troublesome symptom, reported in 60–70% of patients [77,78]. Its pathogenesis is complex, and the results regarding therapy with UDCA showed that it was mostly ineffective in improving this symptom. Since the principal guideline-approved anti-pruritic agents, e.g., cholestyramine, rifampicin, naltrexone, and sertraline, are often ineffective to improve PBC-related pruritus, novel agents targeting this symptom have been developed and are under evaluation.

#### Ileal Bile Acid Transporter (IBAT) Inhibitors

The use of ileal bile acid transporter inhibitors has been suggested for the treatment of PBC-related pruritus due to their ability of decrease retained circulating BAs. IBATs is physiologically devoted to BA reabsorption from the ileum, thus maintaining their enterohepatic circulation. Since in many cholestatic liver diseases, ileal BA absorption is increased, several compounds capable of altering the ileal reabsorption of bile acids have been proposed and are discussed below.

Maralixibat, a selective, sodium-dependent, ileal apical, BA transport inhibitor was tested in a phase 2 trail in which its efficacy and safety were assessed in PBC patients with pruritus (CLARITY study [79]). Patients were divided into arms and treated for 13 weeks with either maralixibat (10 or 20 mg/day) or placebo, in addition to the standard UDCA therapy, when tolerated. The primary endpoint was defined as “adult itch reported outcome average sum score” from baseline to the end of the study. The main ADRs were gastrointestinal disorders, which were common in treated (78.6%) but also placebo (50%) patients. Despite an improvement of baseline risk scores, maralixibat caused no significant improvement of pruritus with respect to placebo.

The IBAT inhibitor GSK2330672, also called linerixibat, was evaluated in a phase 2 trial enrolling 21 patients [80] to assess the safety and tolerability of GSK2330672. The secondary endpoints were changes in patient-reported pruritus scores, assessed by means of different scales, namely a 0 to 10 numerical rating scale, PBC-40 itch domain score, 5-D itch scale, and changes in circulating bile acid levels (NCT01899703). Linerixibat was well tolerated, and diarrhea was the most experienced ADR. The percentage decrease in itch scores was −57% in the numerical rating scale, −31% in the PBC-40 itch domain, and −35% in the 5-D itch scale in linerixibat-treated patients, and these differences were statistically significant with respect to the placebo-treated group. A larger phase 2 study enrolling 147 patients is still ongoing to confirm these beneficial effects on PBS-related pruritus and to further assess the drug tolerability (NCT02966834).

The last proposed IBAT inhibitor, A4250 (odevixibat), was tested in an open-label phase 2 study that aimed to assess drug tolerability and efficacy in improving pruritus in 9 PBC patients (NCT02360852) [81]. Patients were treated with odevixibat at a dose of 0.75 mg (n = 4) or 1.5 mg (n = 5) for 4 weeks. A remarkable improvement in pruritus was observed in all 9 odevixibat-treated patients assessed by VAS, the 5-D itch scale, and the pruritus domain of the PBC-40 questionnaire [82]. Unfortunately, tolerability was low because of gastrointestinal symptoms. Odevixibat received its first approval in the EU in July 2021 for the treatment of progressive familial intrahepatic cholestasis (PFIC) in patients aged ≥ 6 months, followed by its approval in the US for pruritus in patients with PFIC aged ≥ 3 months [83].

### 4.2. Agents Targeting Fatigue

Another frequently reported PBC manifestation is fatigue, a complex syndrome characterized by feelings of discomfort, exhaustion, and lethargy that could significantly reduce the quality of life. The probability of improving fatigue after LT in advanced PBC is roughly 50% [84]. Currently, no pharmacological treatment is approved for PBC-related fatigue. The only prescribed suggestion is an exercise increase, even though this kind of prescription needs further evaluation. The results of a pilot study showed an improvement in muscle pH in PBC patients, and an amelioration of fatigue, social, and emotional symptoms in patients following an exercise program [85].

The first phase 2 randomized controlled trial of treatment of PBS-associated fatigue and daytime somnolence (NCT2376335, [62]) was performed in 57 PBC patients with moderate to severe fatigue. Patients were randomized to receive two doses of rituximab (1000 mg) or placebo. The primary outcome was assessed by measuring fatigue severity using a questionnaire at 3 months. The rationale of the use of rituximab was an improvement in the fatigue associated with a variety of other autoimmune diseases, e.g., Sjogren’s syndrome, which has been also association with PBC. Rituximab, however, failed to show an improvement in fatigue in PBC patients.

Modafinil, an agent acting on the central nervous system and used for the treatment of daytime somnolence in narcoleptic patients, was tested in an open study enrolling 21 patients with PBC experiencing daytime somnolence and fatigue [86]. The starting dose of modafinil was 100 mg/day, which was titrated according to patient’s tolerability and response. Unfortunately, only 14 patients could tolerate the full 2-month treatment, although in those patients, an improvement of excessive daytime somnolence and associated fatigue was observed. The suggestion from these data was to improve the design of the study with a placebo-controlled trial, to confirm modafinil’s efficacy against fatigue.

A preclinical study on an animal model of hepatic cholestasis induced by the ligation of the bile duct demonstrated that early OCA administration was able to improve cognitive impairment [87]. Otherwise, these preclinical observations need to be validated in further studies.

A systematic meta-analysis of 16 studies evaluating UDCA, liver transplantation, serotonin reuptake inhibitors, colchicine, methotrexate, cyclosporine, modafinil, and OCA found some improvement in fatigue with liver transplantation, but a lack of high-quality evidence supporting the efficacy of any other intervention in the treatment of PBC-related fatigue [88].

## 5. Conclusions

In recent years, new candidate drugs have undergone or completed phase 2 and 3 clinical trials on PBC patients who did not respond to the first line therapy with UDCA. OCA represents the most promising drug and is approved in many countries for this indication. Fibrates seem to effectively ameliorate biochemistry alteration and symptoms typical of PBC. Moreover, a variety of new agents, acting with different mechanisms of action, are under clinical evaluations for PBC treatment, e.g., PPAR agonists, NOX inhibitors, immunomodulators, and mesenchymal stem cells transplantation. Even though most of these approaches seem to have beneficial effects on biochemical endpoints, no data are currently available regarding robust endpoints, such as transplant-free survival; thus, their clinical use needs to be supported by the consistent improvement of these parameters. In general, data on the efficacy of the new therapeutic agents are still undergoing investigation in clinical trials and are too premature to provide practical information to physicians. The crucial point when designing clinical trials is the choice of a combination treatment with nuclear receptor ligands and other agents with different mechanism and therapeutic effects [89]. This huge armamentarium of new therapeutic options will likely lead to a novel treatment landscape for PBC in the near future, with novel therapies based on the combinations of multiple agents acting on different pathogenetic mechanisms [90]. Another crucial point is that the ideal therapy for PBC would achieve a complete biochemical remission, namely normalization of serum ALP and bilirubin, and would be well tolerated. Furthermore, the ideal therapy must be safe for patients with advanced or decompensated disease and should aim to reduce the need for liver transplantation [91].

## Figures and Tables

**Figure 1 biomedicines-10-02033-f001:**
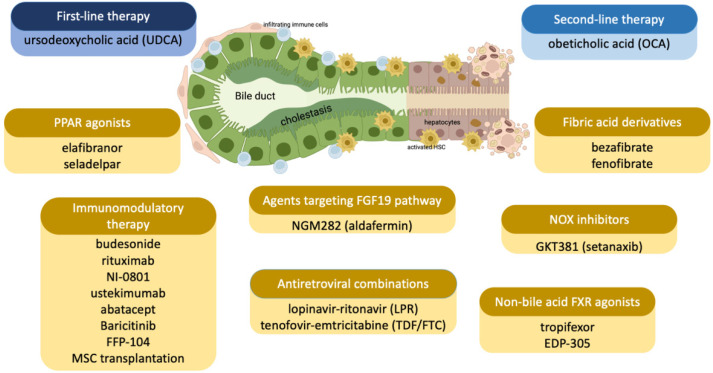
Drugs approved (in blue) or under evaluation (in yellow) for the treatment of PBC.

**Table 1 biomedicines-10-02033-t001:** Ongoing controlled trials with experimental agents in PBC.

Agent	Study Design	Aim/Outcome	Nr. pts	Phase	Duration	NCT nr.
**PPAR agonists**						
BEZA	RCT	Utility of BEZA as add-on therapy/complete biochemical response	34	3	12 m	NCT02937012
FENO	RCT	Clinical efficacy of FENO + UDCA/amelioration of ALP	72	1–2	12 m	NCT02965911
FENO	OL	Utility of FENO + UDCA/complete biochemical response	200	3	44 w	NCT02823353
Seladelpar	OL	Long-term safety and tolerability of seladelpar/measures of adverse events, death	500	3	60 m	NCT03301506
Seladelpar	RCT	Safety and effect of 2 seladelpar regimens on cholestasis/percentage of participants to composite endpoint	240	3	52 w	NCT03602560
BEZA	observational	Influence of BEZA on macrophage activation markers and fibrosis/sCD163 levels	100	3	36 m	NCT04514965
Seladelpar	RCT	Effect of seladelpar on cholestasis/composite endpoint of ALP and total bilirubin	180	3	12 m	NCT04620733
Seladelpar	OL	Effect of hepatic impairment on the pharmacokinetics of seladelpar/pharmacokinetic measures	24	1	17 w	NCT04950764
Saroglitazar Mg	RCT	Safety, tolerability, and efficacy of saroglitazar/improvement in ALP levels	36	2	16 w	NCT03112681
Saroglitazar Mg (EPICS-III)	RCT	Efficacy and safety of saroglitazas/ biochemical response on the composite endpoint of ALP and total bilirubin	192	2 b–3	52 w	NCT05133336
**FXR agonists + PPAR agonists**						
OCA + BEZA	RCT	Effect of OCA + BEZA/change in ALP	75	2	12 w	NCT04594694
OCA + BEZA	RCT	Effect of BEZA alone or in combination with OCA/change in ALP	60	2	12 w	NCT05239468
**FXR Agonists**						
EDP-305	RCT	Safety, tolerability, and efficacy of EDP-305/percentage of participants with at least 20% reduction in ALP	119	2	12 w	NCT03394924
Cilofexor	RCT	Safety and tolerability of cilofexor/percentage of adverse events	71	2	12 w + 30 d	NCT02943447
**Immunomodulants**						
Baricitinib	RCT	Safety and efficacy of baricitinib/change in ALP	52	2	12 w	NCT03742973
MSCs transplantation	RCT	Safety and efficacy of MSC/change in ALP	14	1–2	12 m	NCT03668145
MSCs transplantation	RCT	Safety and efficacy of UC-MSC/change in ALP	100	1–2	12 w	NCT01662973
CNP-104 nanoparticle Incapsulating PDC-E2	RCT	Safety, tolerability, pharmacodynamics of CNP-104 nanoparticle/frequency of adverse events, changes in ALP	40	2	12 d + 20 m	NCT05104853
**Antiretroviral therapy**						
Tenofovir, raltegravir	RCT	Efficacy of antiretroviral therapy/change in ALP	60	2	24 m	NCT03954327

Abbreviations: OCA = obeticholic acid; OL = open label; RCT = randomized controlled trial; MSCs = mesenchymal stem cells; N/A = not available.

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
