# Peer review of "Update on the Pharmacological Treatment of Primary Biliary Cholangitis"

_biomedicines, 2022, doi:10.3390/biomedicines10082033_

Round 1
Reviewer 1 Report
This review study on PBC is well written and organized, and even though it is in line with the topic of the review.
Nevertheless, I think that the shown data on the efficacy of the therapeutic agents listed by the authors and still undergoing investigation in clinical trials are too premature to provide useful information to the readers, or to inspire experts to further a particular line of research in the therapeutic field, in comparison to what is already known.
Author Response
We thank the reviewer for this observation, and we agree with the comment. However, this review aims at updating the most recent advances in the PBC treatment, by focusing on novel drugs or candidate drugs which are currently undergoing clinical evaluations. Even most of them are not approved, the results of the ongoing trials are promising, and the data which have been collected by now, either successful or not, are anyway important for understanding further details of this complex disease and the management of patients. In the revised version, we have improved the conclusion by stressing the fact that most of the therapeutic interventions listed and explained in this Review are currently under investigation for PBC treatment in many clinical trials (highlighted in red). Moreover, we have also improved the conclusion, by adding some suggestions useful for an accurately tuned design of clinical trials, which is surely of outstanding importance for their success. We hope that the Reviewer will appreciate our efforts in rendering this Review more useful and interesting for the Readers.
Reviewer 2 Report
Nice and documented paperwork. Well structured with a lot of studies research. Unfortunately, most of the new drugs intended to use in PBC did not achieve statistical and clinical significance.
The references are extensive and adaptive to the text.
Author Response
We thank the reviewer for these kind comments. This review reported the most recent clinical trials investigating the use of new drugs for PBC patients. Even if most of these drugs are not currently approved, the results of the ongoing trials are promising. Following the reviewer’s suggestion, we have improved the conclusion by stating that many drugs proposed for PBC treatment are currently under investigation and not approved.
Reviewer 3 Report
Floreani A. and co-authors reported a complete an interesting update on current and future therapy for PBC. This review may serve as quick reference for both physician and clinical researchers
I just have some minor comments:
1) Some words should be spent on UDCA mechanism of action in PBC.
2) It would be interesting to have some information or impression on the comparison between UDCA+OCA vs OCA alone, if any.
3) The conclusions section should be expanded including the authors point of view on the possible evolution in therapy in the next decades and the pitfalls in conducting clinical trials in PBC.
4) Line 184: “TIPS replacement..” is maybe “TIPS placement? Please check
Author Response
1. Some words should be spent on UDCA mechanism of action in PBC.
We expanded this point in the text, as suggested.
2. It would be interesting to have some information or impression on the comparison between UDCA+OCA vs OCA alone, if any.
Unfortunately, there are no data on the comparison between UDCA+OCA vs OCA alone. Indeed, more than 95% of patients treated with OCA are also taking UDCA, due to the inclusion criteria (insufficient response to UDCA or intolerance to UDCA). Thus, OCA therapy is in fact an add-on therapy to UDCA.
3. The conclusions section should be expanded including the authors point of view on the possible evolution in therapy in the next decades and the pitfalls in conducting clinical trials in PBC.
We thank the reviewer for the suggestion and improved the conclusion section.
Line 184: “TIPS replacement.” is maybe “TIPS placement? Please check.
We thank the reviewer for this observation and modified the text accordingly.
Round 2
Reviewer 1 Report
I suggest to include in table 1 the aims of study and outcomes related to the various drugs.
Author Response
We thank the Reviewer for this suggestion. We did not add aims and outcomes in the previous version, because this is uncommon for ongoing studies. However, in order to comply with the Reviewer’s request, we add this information to the Table summarizing ongoing studies.
We hope that this addition sufficiently improves the quality of our manuscript and that it is now suitable for publication.
Round 3
Reviewer 1 Report
Dear authors, despite the limitations already mentioned, the paper is better organized in its current form.